# First cases of SARS-CoV-2 infection and secondary transmission in Kisumu, Kenya

**Beth A. Tippett Barr**[1,2☯]*, **Amy Herman-Roloff**[1☯], **Margaret Mburu**[3], **Pamela M. Murnane**[4], **Norton Sang**[5], **Elizabeth Bukusi**[5], **Elizabeth Oele**[6], **Albert Odhiambo**[6], **Jayne Lewis-Kulzer**[4], **Clayton O. Onyango**[1], **Elizabeth Hunsperger**[1], **Francesca Odhiambo**[5], **Rachel H. Joseph**[1], **Peninah Munyua**[1], **Kephas Othieno**[3], **Edwin Mulwa**[5], **Victor Akelo**[1], **Erick Muok**[3], **Marc Bulterys**[1], **Charles Nzioka**[7], **Craig R. Cohen**[4]

1 U.S. Centers for Disease Control and Prevention, Kisumu, Kenya, 2 Nyanja Health Research Institute, Salima, Malawi, 3 Kenya Medical Research Institute, Kisumu, Kenya, 4 University of California, San Francisco, San Francisco, CA, United States of America, 5 Kenya Medical Research Institute, Nairobi, Kenya, 6 Kisumu County Department of Health, Kisumu, Kenya, 7 Ministry of Health, Nairobi, Kenya

☯ These authors contributed equally to this work.
* Beth@nyanja-health.com

**Data Availability Statement:** All data are in the manuscript and/or supporting information files.

**Funding:** This investigation was funded through a grant from the CDC Foundation to the University of

## Abstract

We investigated the first 152 laboratory-confirmed SARS-CoV-2 cases (125 primary and 27 secondary) and their 248 close contacts in Kisumu County, Kenya. Conducted June 10–October 8, 2020, this study included interviews and sample collection at enrolment and 14–21 days later. Median age was 35 years (IQR 28–44); 69.0% reported COVID-19 related symptoms, most commonly cough (60.0%), headache (55.2%), fever (53.3%) and loss of taste or smell (43.8%). One in five were hospitalized, 34.4% >25 years of age had at least one comorbidity, and all deaths had comorbidities. Adults ≥25 years with a comorbidity were 3.15 (95% CI 1.37–7.26) times more likely to have been hospitalized or died than participants without a comorbidity. Infectious comorbidities included HIV, tuberculosis, and malaria, but no current cases of influenza, respiratory syncytial virus, dengue fever, leptospirosis or chikungunya were identified. Thirteen (10.4%) of the 125 primary infections transmitted COVID-19 to 27 close contacts, 158 (63.7%) of whom resided or worked within the same household. Thirty-one percent (4 of 13) of those who transmitted COVID-19 to secondary cases were health care workers; no known secondary transmissions occurred between health care workers. This rapid assessment early in the course of the COVID-19 pandemic identified some context-specific characteristics which conflicted with the national line-listing of cases, and which have been substantiated in the year since. These included over two-thirds of cases reporting the development of symptoms during the two weeks after diagnosis, compared to the 7% of cases reported nationally; over half of cases reporting headaches, and nearly half of all cases reporting loss of taste and smell, none of which were reported at the time by the World Health Organization to be common symptoms. This study highlights the importance of rapid in-depth assessments of outbreaks in understanding the local epidemiology and response measures required.

California San Francisco (PMM, JLK, CRC). The opinions expressed by authors contributing to this manuscript do not necessarily reflect the opinions of the CDC Foundation, the U.S. Centers for Disease Control and Prevention, or the institutions with which the authors are affiliated. The funders had no role in study design, data collection and analysis, decision to publish, or preparation of the manuscript.

**Competing interests:** The authors have declared that no competing interests exist.

## Introduction

Coronaviruses are common, and typically cause mild disease with cold symptoms; however, over the past two decades, severe acute respiratory syndrome coronavirus 1 (SARS-CoV-1) and Middle East respiratory syndrome coronavirus (MERS-CoV) have caused severe morbidity and mortality [1]. In late 2019, a novel beta-coronavirus, SARS-CoV-2, was first reported in Wuhan, China [2]. Infection with SARS-CoV-2 can result in Coronavirus disease 2019 (COVID-19) which was declared a public health emergency of international concern by the World Health Organization (WHO) on January 30, 2020 and a global pandemic on March 11, 2020 [3]. Kenya reported its first case of laboratory-confirmed COVID-19 on March 12, 2020 [4].

Given that SARS-CoV-2 was a novel virus, the detection and spread of this emerging pathogen was accompanied by epidemiological, clinical and virological questions–especially given its transmissibility and pathogenicity. WHO developed a template protocol for the First Few 'X' cases (FFX) investigation of COVID-19 cases and their close contacts to improve understanding of SARS-CoV-2 transmission, clinical progression, and risk factors, and to guide response measures, ideally starting with the first case [5]. Nigeria [6] in West Africa and Zambia [7] in Southern Africa have reported on the initial COVID-19 in their countries; here we present the first FFX results from Kenya, in East Africa.

This FFX investigation was conducted in Kisumu County in western Kenya. With a population of 1.1 million, Kisumu County is the twelfth most populous of the 47 counties in Kenya, and Kisumu City is the third largest urban center and one of the largest inland port cities on Lake Victoria. Kisumu City is connected by a highway to the Kenya-Uganda border to the west and to the capital, Nairobi, and costal port city, Mombasa, to the southeast. The Kisumu County Department of Health (KCDoH) overseas the public health program within the county with a special focus on human immunodeficiency virus (HIV), tuberculosis, and malaria–all substantial causes of morbidity in the County [8]. CDC Kenya and the Kenya Medical Research Institute (KEMRI) operate a high-capacity reference laboratory in Kisumu County, which performed nearly 90% of SARS-CoV-2 testing in the county and close to 20% of all testing nationally by the end of 2020.

The first laboratory-confirmed COVID-19 case in Kisumu County was identified on June 9, 2020. During this period, 40,178 cases were identified in Kenya; the majority occurring in Nairobi and Mombasa counties. National reporting indicated that nearly all COVID-19 infections had resulted from local transmission (98%) and the majority were asymptomatic (93%). During the first wave of COVID-19 in 2020 which peaked in July 2020, cough (51%) and fever (35%) were the most common symptoms reported, and the case fatality rate was 1.9% [9].

This investigation of the first few cases and their contacts in Kenya was designed to increase the understanding of the spread of SARS-CoV-2, describe clinical progression in the presence of comorbid conditions and inform local public health strategies to reduce transmission and improve clinical management.

## Methods

### Ethics statement

The investigation protocol received a non-research determination from the U.S. Centers for Disease Control and Prevention (#NCIRD-IE-2/12/20-d2ff4), and approval from the Kenya M Medical Research Institute's Scientific Ethics Review Unit (#P00138/4037) and the University of California San Francisco (UCSF) Institutional Review Board (#280847). Formal written consent was obtained from each study participant aged 18 years and above, and consent was obtained from parents for study participants aged below 18 years.

## Study design

This was a prospective study of laboratory-confirmed COVID-19 cases and their identified close contacts from June to October 2020 conducted in coordination with the KCDoH's outbreak response (Fig 1). Index cases were identified through existing surveillance systems as those with a) laboratory-confirmed COVID-19 infection and b) self-reported residence in Kisumu County.

Over 90% of cases were identified through the Kenya Medical Research Institute's Center for Global Health Research (KEMRI-CGHR) laboratory, which was one of the few laboratories with capacity to test for SARS-CoV-2 at the time. The few cases reported by private laboratories were retested at KEMRI-CGHR to ensure diagnosis congruency.

Following laboratory diagnosis and reporting, initial case patient visits and contact identifications were conducted in coordination with KCDoH's outbreak, observing the Ministry of Health's (MoH) procedures and guidance on quarantine, isolation, and infection, prevention and control (IPC) measures. All participants aged 18 and above voluntarily provided written consent for themselves (ages 18+), witnessed by the study data collector and Ministry of Health outbreak response personnel present. Children aged 13–17 provided written assent in addition

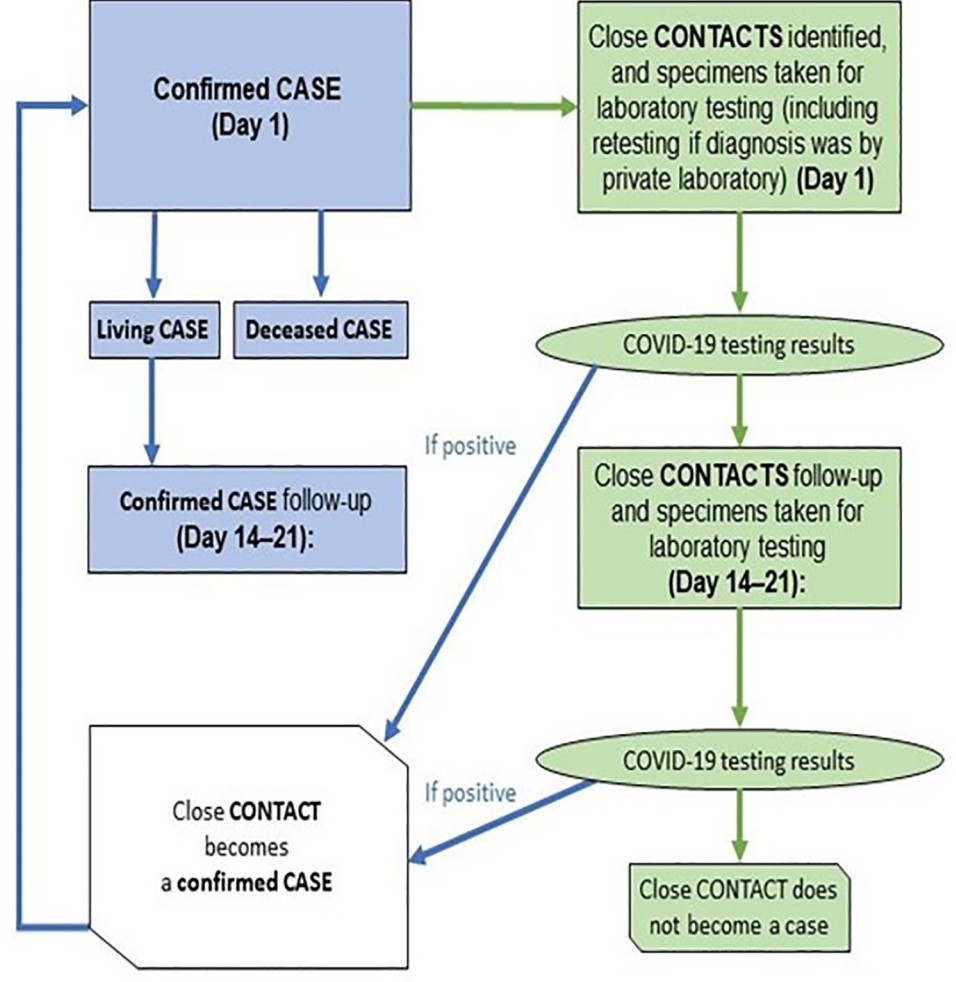

**Fig 1. Study flow for the first few cases of COVID-19 in Kisumu, Kenya, June-October 2020.**

to written parental consent. In the event of death, next of kin consent was identified to provide consent to include the deceased in the study. A standardized questionnaire was administered by trained FFX and KCDoH staff to participants to establish baseline clinical characteristics, and to identify all eligible close contacts. The investigation questionnaires were based on WHO proposed forms [7] and adapted to capture important communicable and non-communicable causes of morbidity and mortality in Kisumu County. Data were abstracted from clinical records for hospitalized patients. Data collection for the study was done in addition to data collection for routine public health monitoring.

Close contacts were defined as individuals who were reported by an index case as someone they had spent over 15 minutes with, within 2 meters, between 48 hours prior to symptoms through the time of the enrollment interview. Only contacts residing in Kisumu County were enrolled in the study. Others were recorded for the KCDoH public health response, but not invited to participate. At the start of this study, mask mandates were not yet implemented, and it is therefore assumed nobody except health care providers in the work setting were wearing masks.

All contacts were tested and advised on quarantine procedures by KCDoH staff according to Kenya MoH guidelines. Contacts with laboratory-confirmed COVID-19 infection were followed-up by KCDoH (public health response) and FFX investigation staff (study procedures) together. For each enrolled participant (cases and close contacts), an initial data and specimen collection visit occurred on day one, and again 14–21 days after enrolment (Fig 1).

A nasopharyngeal swab, oropharyngeal swab, and venous blood sample were collected from all cases on the day of enrolment. Whole blood and serum were collected for antigen-based, molecular and immunological testing for endemic infectious diseases in this region including malaria, human immunodeficiency virus (HIV), tuberculosis (TB), malaria, dengue, chikungunya, leptospirosis, influenza A and respiratory syncytial virus (RSV). Follow-up samples included upper respiratory tract samples and clotted blood, and all cases were offered rapid tests for HIV, malaria, and pregnancy (for women of reproductive age) during home visits. HIV testing and counselling were offered per the Kenya MoH guidelines and those newly diagnosed with HIV or pregnant were referred to care. Those testing positive for malaria were offered treatment immediately, with referral to a facility if not rapidly resolving. Depending on the series of tests that the participant consented or assented to, 5-20mL of blood was collected.

Overweight and obesity was both self-reported and assessed by study staff using a visual scale. Concurrency between the two was >90% and in the event of non-concurrence, the study staff assessment was used for analysis. Other non-communicable morbidities (e.g., diabetes) were self-reported and not clinically confirmed.

## Data collection

Data were captured on tablets using KoBo Toolbox (Harvard Humanitarian Initiative, Boston, USA), an open-source electronic field data collection tool utilizing secure local servers and a cloud-based back-up system. The study used the same unique identification number assigned to cases by the KCDoH in the course of routine COVID-19 case investigations in order to align with on-going public health response efforts.

## Statistical analysis

We used STATA Version 15 (StataCorp, College Station, USA) to conduct descriptive analyses of the cases (S1 Data) and contacts (S2 Data). Mean (standard deviation [SD]) was used for normally distributed continuous variables and median (interquartile range [IQR]) for non-normally distributed continuous variables. For categorical variables, we report counts and

proportions. The variables included in this analysis included age, gender, occupation, activities prior to study enrolment, COVID-19 symptoms and severity of the disease, and comorbidities. Severe disease was defined as requiring hospitalization or resulting in death. Age groups were grouped by perceived risk of poor outcomes at the early stage in the pandemic: Children and young adults at lowest risk (<25 years of age) and oldest adults at highest risk (>55 years of age). For this reason, analysis on the association between infectious and chronic comorbidities with COVID-19 disease severity was restricted to adults only. Graphical representation to show the weekly trend of reported occupations among cases was done in Microsoft Excel, and that of secondary transmission dynamics were developed using Microsoft PowerPoint.

# Results

## COVID-19 cases

Of the first 152 cases of COVID-19 in Kisumu, Kenya, 125 (82.2%) were primary cases; 27 (17.8%) were secondary cases in contacts, transmitted by 12 (9.6%) of the primary cases. In all 152 cases, children and young adults (<25 years) represented 11.2% (10 of 89) of male and 22.2% (14 of 63) of female cases; this age group comprises 45.2% (57 of 126) of male and 59.8% (73 of 122) of female contacts identified [Table 1]. Transport drivers made up nearly one quarter of male cases (20 of 89). Over one third of cases (38.8%, 59 of 152) were office workers. The most common activities reported by cases in the 14 days prior to COVID-19 diagnosis were use of public transport (53.3%, 81 of 152), grocery shopping (58 of 152, 38.2%) and visiting a health facility (47 of 152, 30.9%).

The outbreak in Kisumu started with cases occurring in transport drivers, including international cargo drivers and local public transport drivers (Fig 2), transitioning within a few weeks to office workers. The subsequent sustained community transmission resulted in an uptick in cases among students and young children (schools were closed at the time), those with informal employment.

Overall, 31 (20.4%) cases were hospitalized (Table 2), 4 (12.9%) of whom were admitted to the intensive care unit (ICU) and survived while 3 (9.7%) died (CFR 2.0% [95% CI 0.41%-5.66%]). Two thirds of cases reported experiencing any symptoms at enrolment or during the two-week follow-up period; symptomatic infection was most common in those aged ≥55 years (92.3%), and least common in those aged <25 years (45.8%). Cough (60.0%), headache (55.2%), fever (53.3%), loss of appetite (47.6%), fatigue (44.8%), and loss of smell/taste (43.8%) were the most common symptoms reported, with variation by age.

Comorbidity increased with age: children and young adults under age 25 had a low risk profile, with only 1 (4.2%) overweight, 1 (4.2%) with asthma, and 1 (4.2%) with HIV and on antiretroviral treatment (ART) (Table 2). The prevalence of self-reported comorbidities increased with age; in adults aged >55, two thirds (69.2%) of had at least one comorbidity, and over half (53.9%) had diabetes. Nearly one quarter (22.7%) of all adults >25 were categorized as overweight or obese. Overall HIV prevalence was 11.2% and increased with age from 4.2% in those under 25 to 15.4% in those 55 and above. Of the 14 adults between the ages of 25 and 54 who self-reported a previous HIV diagnosis, three (21.4%) reported not currently being on ART. Only one HIV-infected adult required intensive care (6.3%), four (25.0%) were hospitalized but not in need of intensive care (25.0%), and the remainder (68.8%) were successfully managed as outpatients.

Additional serology testing for the presence of infectious diseases returned multiple positive results, including RSV: 23 (16.0%) respiratory syncytial virus, 18 (12.5%) Chikungunya: 18 (12.5%), 8 (5.6%) Influenza A, 4 (3.5% Leptospira, and 1 (0.7%) Dengue. Real-time PCR for

**Table 1. Demographic profile of COVID-19 cases and contacts, FFX study, Kisumu, Kenya, June-October 2020.**

| | Characteristics | Cases n(%) | | | Contacts n(%) | | |
|---|---|---|---|---|---|---|---|
| | | Male (n = 89) | Female (n = 63) | Total (n = 152) | Male (n = 126) | Female (n = 122) | Total (n = 248) |
| Age group (years) | <15 | 8 (44.4) | 10 (55.6) | 18 | 38 (45.2) | 46 (54.8) | 84 |
| | 15–24 | 2 (33.3) | 4 (66.7) | 6 | 19 (41.3) | 27 (58.7) | 46 |
| | 25–34 | 27 (51.9) | 25 (48.1) | 52 | 29 (54.7) | 24 (45.3) | 53 |
| | 35–44 | 29 (69.1) | 13 (31.0) | 42 | 26 (68.4) | 12 (31.6) | 38 |
| | 45–54 | 13 (61.9) | 8 (38.1) | 21 | 10 (71.4) | 4 (28.6) | 14 |
| | 55+ | 10 (76.9) | 3 (23.1) | 13 | 4 (30.8) | 9 (69.2) | 13 |
| Occupation | Office Workers | 33 (55.9) | 26 (44.1) | 59 | 43 (72.9) | 16 (27.1) | 59 |
| | Transport drivers | 20 (100) | 0 (0.0) | 20 | 5 (100) | 0 (0.0) | 5 |
| | *Trucks* | *14 (70.0)* | *0 (0.0)* | *14* | *1 (20.0)* | *0 (0.0)* | *1* |
| | *Public Transport* | *6 (30.0)* | *0 (0.0)* | *6* | *4 (80.0)* | *0 (0.0)* | *4* |
| | Health Care Workers | 7 (43.8) | 9 (56.3) | 16 | 7 (38.9) | 11 (61.1) | 18 |
| | Students and preschoolers | 10 (43.5) | 13 (56.5) | 23 | 57 (46.3) | 66 (53.7) | 123 |
| | Other | 22 (57.9) | 16 (42.1) | 38 | 16 (34.8) | 30 (65.2) | 46 |
| Activities during 14 days prior to study enrolment | Visited health facility | 29 (61.7) | 18 (38.3) | 47 | 13 (32.5) | 27 (67.5) | 40 |
| | Used public transport | 42 (51.9) | 39 (48.1) | 81 | 42 (56.8) | 32 (43.2) | 74 |
| | Went to workplace | 31 (70.5) | 13 (29.5) | 44 | 28 (63.6) | 16 (36.4) | 44 |
| | Went grocery shopping | 30 (51.7) | 28 (48.3) | 58 | 53 (41.7) | 74 (58.3) | 127 |
| | Attended any gathering | 18 (64.3) | 10 (35.7) | 28 | 26 (39.4) | 40 (60.6) | 66 |
| | No travel or activities | 15 (46.9) | 17 (53.1) | 32 | 44 (55.7) | 35 (44.3) | 79 |

RSV, Influenza-A, Chikungunya, Leptospira and Dengue on samples that were IgM positive returned negative for these pathogens.

Hospitalization was associated with chronic but not infectious comorbidity (Table 3): There was strong evidence of an association between previous diabetes diagnosis and COVID-19 hospitalization (OR 7.5, 95%CI:2.3–24.7, p<0.001), but not between overweight or obesity and hospitalization (OR 1.7, 95%CI:0.7–4.1, p = 0.28). Seventeen percent of adults with no comorbidity were hospitalized compared to 39% of adults with any comorbidity (OR 3.2, 95% CI:1.4–7.3, p = 0.01), and over half of those with two or more comorbidities were hospitalized (OR 6.4, 95%CI: 2.1–20.2, p<0.001).

Of the 31 (20%) cases hospitalized, 6 were admitted to the ICU, 2 of whom died (Fig 3); one additional death occurred at home after discharge from the hospital. All six cases admitted to ICU who recovered received supplemental oxygen and all had secondary bacterial infections; two were diagnosed with pneumonia. In those who died, one had systemic inflammatory response syndrome, one had hypotension requiring vasopressors, and one had ARDS. Nearly half of those hospitalized without ICU admission (11 of 25, 44%) experienced secondary bacterial infections.

## COVID-19 Contacts and secondary transmission

Of the first 152 cases in Kisumu, 27 (17.9%) were secondary cases, transmitted by 13 (10.4%) of the 125 primary cases. Of the primary cases who were known to transmit to close contacts, Fig 4 shows that healthcare workers infected as a primary case did not transmit to the health care worker peers. The average number of symptoms in primary cases who transmitted

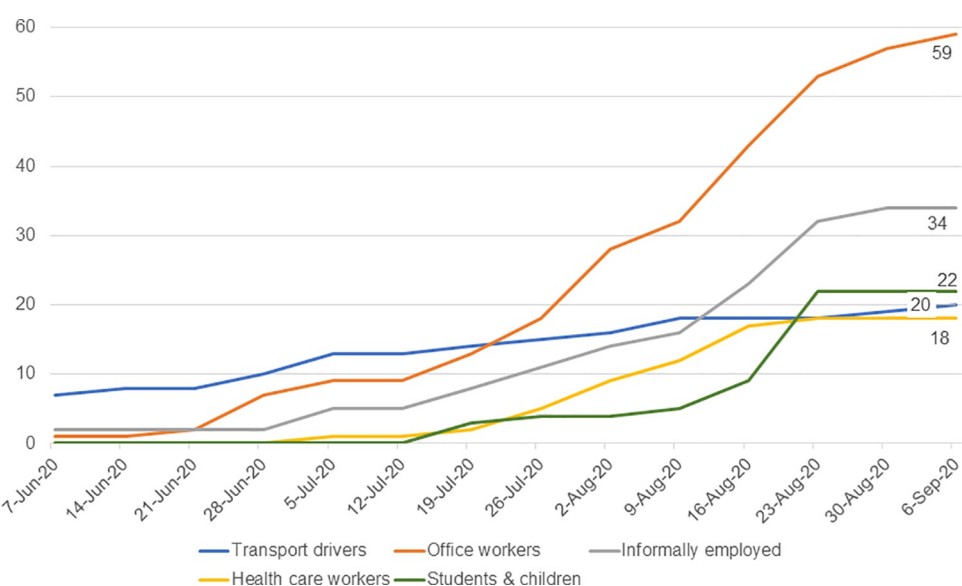

**Fig 2. Weekly occupation trends among cases, FFX study, Kisumu, Kenya (June-October 2020).**

SARS-CoV-2 to others was 20 percent higher than the average number of symptoms in those cases not documented to have transmitted the virus.

Across the reported contacts, there was no evidence of association between SARS-CoV-2 acquisition and age, sex, chronic or infectious comorbidity, or self-reported social gatherings and travel (Table 4).

## Discussion

This investigation was designed to increase our understanding of COVID-19 transmission and explore clinical progression in the presence of other highly endemic diseases in Kisumu County. During a time of extensive lockdown in Kenya where public movement was restricted, it is logical that the first cases appear to have been imported by truck drivers, and then local transmission occurred in office settings, followed by broader community transmission. Secondary transmission was lower than expected. Health care workers comprised 31% of the participants who transmitted COVID-19 to secondary cases; however, no secondary transmissions were amongst their coworker contacts, indicating these infection prevention measures were effective. Mask mandates for the general public became mandatory during the period of study implementation, although it was observed by the authors that masks were inconsistently and incorrectly worn, often with nose and mouth left exposed. It is not believed that masks were worn in home settings.

In this sample, two thirds of cases reported symptoms during the first two weeks after diagnosis, but the burden of most common symptoms (cough, headache and fever) were not in the same order as those reported by WHO (cough, fever and fatigue) [10]. Over half of all participants reported experiencing headache, making it the second most prevalent symptom across all age groups; although not a commonly reported symptom globally at the time [10]. Among children, COVID-19 presented with symptoms similar to a common cold. This finding underscores the necessity of conducting high quality surveillance in the local context to understand how emerging diseases may present differently in different populations, and how to tailor interventions for the greatest impact.

**Table 2. Symptoms, comorbidities and severity of disease by age group, FFX study, Kisumu, Kenya, June-October 2020.**

| | | Age Group (n/%) | | | |
| --- | --- | --- | --- | --- | --- |
| | | <25 | 25–54 | 55+ | Total |
| | | (n = 24) | (n = 115) | (n = 13) | (n = 152) |
| Severity of COVID-19 | Not hospitalized | 24 (100.0) | 92 (80.0) | 5 (38.5) | 121 (79.6) |
| | Hospitalized but no ICU | 0 (0.0) | 18 (15.7) | 6 (46.2) | 24 (15.8) |
| | ICU without death | 0 (0.0) | 3 (2.6) | 1 (7.7) | 4 (2.6) |
| | Death | 0 (0.0) | 2 (1.7) | 1 (7.7) | 3 (2.0) |
| Symptom prevalence | Asymptomatic | 11 (45.8) | 35 (30.4) | 1 (7.7) | 47 (31.6%) |
| | Any symptom | 13 (54.2) | 80 (69.6) | 12 (92.3) | 105 (69.0) |
| | *1–3 symptoms* | *10 (41.7)* | *13 (16.3)* | *4 (30.8)* | *38 (25.7%)* |
| | *4+ symptoms* | *3 (12.5)* | *56 (48.7)* | *8 (61.5)* | *67 (43.4%)* |
| | Fever or cough or fatigue* | 6 (25.0) | 64 (55.7) | 9 (69.2) | 79 (52.0%) |
| Specific symptoms | Cough | 6 (25.0) | 49 (42.6) | 8 (66.7) | 63 (60.0) |
| | Headache | 4 (16.7) | 49 (42.6) | 5 (41.7) | 58 (55.2) |
| | Fever / chills | 4 (16.7) | 46 (40.0) | 6 (50.0) | 56 (53.3) |
| | Loss of appetite | 4 (16.7) | 38 (33.0) | 8 (66.7) | 50 (47.6) |
| | Fatigue | 1 (4.2) | 40 (34.8) | 6 (50.0) | 47 (44.8) |
| | Loss of smell/taste | 5 (20.1) | 38 (33.0) | 3 (25.0) | 46 (43.8) |
| | Sore throat | 3 (12.5) | 34 (29.6) | 5 (41.7) | 42 (40.0) |
| | Muscle/Joint aches | 2 (8.3) | 28 (24.3) | 7 (58.3) | 37 (35.2) |
| | Runny nose | 6 (25.0) | 23 (20.0) | 3 (25.0) | 32 (30.5) |
| | Shortness of breath | 0 (0.0) | 22 (19.1) | 2 (16.7) | 24 (22.9) |
| | Diarrhea | 4 (16.7) | 19 (16.5) | 1 (8.3) | 24 (22.9) |
| | Other symptoms | 1 (4.2) | 12 (10.4) | 3 (25.0) | 16 (15.2) |
| Comorbidity prevalence | None | 21 (87.5) | 80 (69.6) | 4 (30.8) | 105 (69.1) |
| | Any comorbidity | 3 (12.5) | 35 (30.4) | 9 (69.2) | 47 (30.9) |
| | *1 only* | *2 (66.7)* | *25 (71.4)* | *3 (33.3)* | *30 (63.8)* |
| | *2+* | *1 (33.3)* | *10 (28.6)* | *6 (66.7)* | *17 (36.2)* |
| Non-communicable comorbidities and risk factors | Underweight | 0 (0.0) | 1 (1.9) | 0 (0.0) | 1 (0.7) |
| | Overweight/Obese | 1 (4.2) | 26 (22.6) | 3 (23.1) | 30 (19.7) |
| | Diabetes | 0 (0.0) | 7 (6.1) | 7 (53.9) | 14 (9.2) |
| | Asthma | 1 (4.2) | 4 (3.5) | 0 (0.0) | 3 (2.0) |
| | Heart Disease | 0 (0.0) | 1 (0.9) | 1 (7.7) | 2 (1.3) |
| | Chronic lung disease | 0 (0.0) | 2 (1.7) | 0 (0.0) | 0 (0.0) |
| | Cancer | 0 (0.0) | 0 (0.0) | 1 (7.7) | 1 (0.7) |
| | Other | 0 (0.0) | 13 (34.3) | 5 (38.5) | 20 (13.2) |
| Infectious comorbidities** | HIV | 1 (4.2) | 14 (12.2) | 2 (15.4) | 17 (11.2) |
| | *Newly diagnosed* | *0 (0.0)* | *1 (6.7)* | *0 (0.0)* | *1 (5.9)* |
| | *On ART* | *1 (100)* | *10 (71.4)* | *2 (100)* | *13 (76.5)* |
| | *Not on ART* | *0 (0.0)* | *3 (21.4)* | *0 (0.0)* | *3 (17.6)* |
| | Tuberculosis | 1 (4.2) | 0 (0.0) | 1 (7.7) | 1 (0.7) |
| | Malaria | 0 (0.0) | 3 (0.0) | 0 (0.0) | 3 (0.0) |

*Most common symptoms of COVID-19 reported globally (https://www.who.int/emergencies/diseases/novel-coronavirus-2019)

**Further testing for the presence of infectious diseases including Leptospirosis, Dengue, Influenza, Respiratory Syncytial Virus (RSV) and Chikungunya all tested negative by polymerase chain reaction (PCR) testing

**Table 3. Comorbidity and disease severity in adults aged ≥25 years, FFX study, Kisumu, Kenya (June-October 2020).**

| | | No Hosp (n = 97) | Hosp (n = 31) | | p-value | Total (n = 128) |
|---|---|---|---|---|---|---|
| | | | | OR (95% CI) | | |
| Comorbidity prevalence | None | 70 (88.1) | 14 (16.7) | Ref | | 84 (100) |
| | Any comorbidity | 27 (61.4) | *17 (38.6)* | **3.2 (1.4–7.3)** | **0.01** | *44 (100)* |
| | *1 only* | *20 (71.4)* | *8 (28.6)* | *2.0 (0.7–5.4)* | *0.16* | *28 (63.6)* |
| | *2+* | *7 (43.8)* | *9 (56.2)* | **6.4 (2.1–20.2)** | **<0.001** | *16 (36.4)* |
| Selected non-communicable comorbidities and risk factors | Underweight | 2 (100.0) | 0 (0.0) | - | | 2 (100) |
| | Overweight/obese | 21 (67.7) | 10 (32.3) | 1.7 (0.7–4.1) | 0.28 | 31 (100) |
| | Any abnormal weight | 23 (70.0) | 10 (30.0) | 1.5 (0.6–3.7) | 0.35 | 33 (100) |
| | Diabetes | 5 (35.7) | 9 (64.3) | **7.5 (2.3–24.7)** | **<0.001** | 14 (100) |
| | Asthma | 3 (75.0) | 1 (25.0) | 1.0 (0.1–10.4) | 0.97 | 4 (100) |
| | Heart Disease | 2 (100.0) | 0 (0.0) | - | | 2 (100) |
| | Cancer | 0 (0.0) | 1 (100.0) | - | | 1 (100) |
| Infectious comorbidities | HIV | 11 (68.8) | 5 (31.3) | 1.5 (0.5–4.7) | 0.49 | 16 (100) |
| | *HIV+, ID'd in study* | *1 (100.0)* | *0 (0.0)* | *-* | | *1 (6.3)* |
| | *HIV+ on ART* | *8 (66.7)* | *4 (33.3)* | *1.7 (0.5–5.9)* | *0.44* | *12 (75.0)* |
| | *HIV+ not on ART* | *2 (67.7)* | *1 (33.3)* | *1.77 (0.1–19.0)* | *0.69* | *3 (18.8)* |
| | Tuberculosis | 1 (100.0) | 0 (0.0) | - | | 1 (100) |
| | Malaria | 2 (67.7) | 1 (33.3) | 1.58 (0.1–18.1) | 0.71 | 3 (100) |

\*Any case who died or was admitted to hospital, regardless of the extent of clinical care required

In this investigation, in addition to the symptom profile differing from global reports, the prevalence of symptoms reported nationally (7%) [9] was only one tenth of that reported by participants in this study (70%). This is likely a result of both extensive questioning of participants on symptoms, as well as the inclusion of a 14-day follow-up period in the study. The national estimate used only those symptoms reported at the time of sample collection for initial diagnosis, where this study documented symptoms both at diagnoses and two weeks later. The proportion of symptomatic infections in our study was similar to meta-analysis estimates that 54% of cases reported globally were symptomatic [11]. Some studies assert that asymptomatic SARS-CoV-2 infection is rare–occurring only in 5% of cases [12]. To some extent, it is

- 27-year old male with pre-existing asthma who developed acute respiratory distress syndrome (ARDS), pneumonia confirmed by chest X-ray, a secondary bacterial infection, and hypotension requiring vasopressors. He died while receiving supplemental oxygen and mechanical ventilation.
- 35-year old male with diabetes and hypertension. He had pneumonia on chest x-ray and died while receiving supplemental oxygen.
- 59-year old male with diabetes, hypertension, and a blood infection. He was tested for COVID-19 when diagnosed with pneumonia. He was transferred to a hospital for mechanical ventilation, improved, tested negative for SARS-CoV-2, and was then transferred back to the original admitting hospital. They in turn could not admit him without his COVID-negative results available, which took 24 hours to secure while he went home. The next day he was admitted to hospital again. His condition deteriorated and he was transferred to ICU and died there.

**Fig 3. Summary of COVID-19 deaths in Kisumu, FFX Study, Kenya, June-October 2020.**

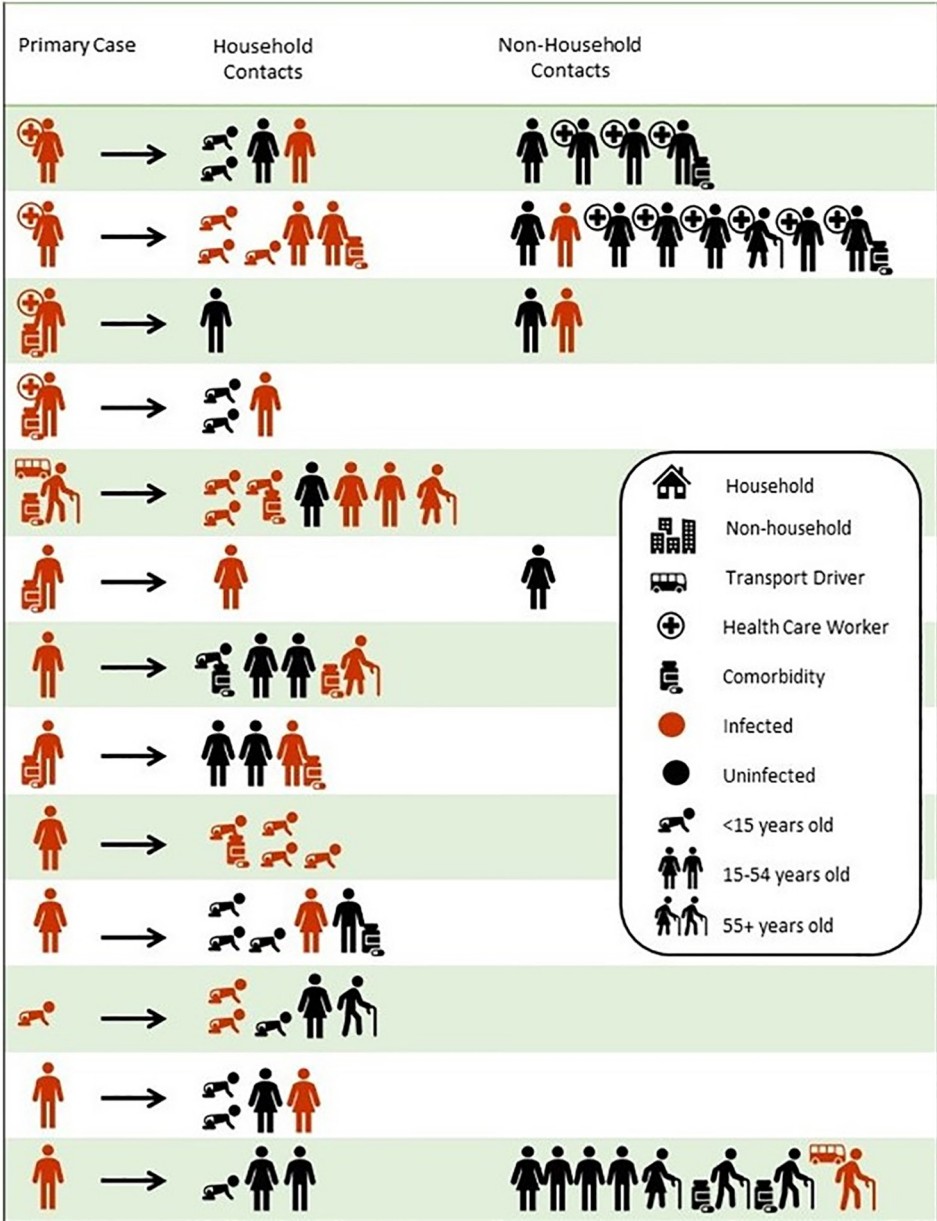

**Fig 4. Visual Representation of Secondary Transmission, FFX Study, Kisumu, Kenya, June-October 2020.**

difficult to compare the proportion of symptomatic infection collected as a part of a study with data collected as part of a public health response when the resources dedicated to the collection of these data differed considerably (e.g., active data collection across multiple time points compared to convenience follow-up in a resource-constrained health system). However, this large discrepancy between early reports of symptom prevalence in the national level surveillance system and that documented in this study raise the importance of integrating follow-up into early surveillance efforts of new pathogens to ensure disease progression is appropriately documented, reported and factored into prevention and control measures.

**Table 4. Risks associated with acquiring COVID-19 infection in Contacts, FFX study, Kisumu, Kenya, June-October 2020.**

| | | Contacts who stayed negative N = 221 (89.1%) | Contacts who became cases N = 27 (10.9%) | OR (95% CI) | p-value | Total N = 248 |
|---|---|---|---|---|---|---|
| Age group (years) | <25 | 116 (89.2) | 14 (10.8) | ref | | 130 (52.4) |
| | 25–54 | 94 (89.5) | 11 (10.5) | 1.0 (0.4–2.2) | 0.94 | 105 (42.3) |
| | 55+ | 11 (84.6) | 2 (15.4) | 1.5 (0.3–7.5) | 0.62 | 13 (5.2) |
| Sex | Male | 114 (90.5) | 12 (9.5) | Ref | | 126 (50.8) |
| | Female | 107 (87.7) | 15 (12.3) | 1.3 (0.6–3.0) | 0.49 | 122 (49.2) |
| | | | | | | N = 245 |
| Comorbidity | None | 184 (88.9) | 23 (11.1) | Ref | | 207 (84.5) |
| | Any infectious or chronic comorbidity | 35 (92.1) | 3 (7.9) | 0.7 (0.2–2.4) | 0.56 | 38 (15.5) |
| Non-communicable comorbidities and risk factors | Abnormal weight | 30 (88.2) | 4 (11.8) | 1.2 (0.4–3.6) | 0.81 | 34 (13.9) |
| | Diabetes | 3 (75.0) | 1 (25.0) | 2.9 (0.3–28.6) | 0.37 | 4 (1.6) |
| | Asthma | 12 (92.3) | 1 (7.7) | 0.7 (0.1–5.8) | 0.75 | 13 (5.3) |
| | Heart Disease | 2 (100) | 0 (0.0) | - | | 2 (0.8) |
| | Chronic lung disease | 3 (100) | 0 (0.0) | - | | 3 (1.2) |
| | Cancer | 3 (100) | 0 (0.0) | - | | 3 (1.2) |
| Infectious comorbidities | HIV | 13 (92.9) | 1 (7.1) | 0.7 (0.1–5.5) | 0.71 | 14 (5.7) |
| | Tuberculosis | 6 (85.7) | 1 (14.3) | 1.4 (0.2–12.3) | 0.75 | 7 (2.9) |
| | Malaria | 5 (83.3) | 1 (16.7) | 1.5 (0.2–13.2) | 0.74 | 6 (2.4) |
| Occupation | Office Workers | 52 (89.7) | 6 (10.3) | 1.0 (0.4–2.4) | 0.88 | 58 (23.7) |
| | Transport drivers | 5 (100) | 0 (0.0) | - | | 5 (2.0) |
| | Health Care Workers | 18 (100) | 0 (0.0) | - | | 18 (7.3) |
| | Students and preschoolers | 108 (88.5) | 14 (11.5) | 1.1 (0.5–2.5) | 0.77 | 122 (49.8) |
| | Other Occupation | 38 (84.4) | 7 (15.6) | 1.7 (0.7–4.3) | 0.27 | 45 (18.4) |
| Social risk | Attended gathering or traveled | 149 (88.2) | 20 (11.8) | 1.4 (0.6–3.4) | 0.49 | 169 (69.0) |

In this investigation, nearly half (43.8%) of participants lost their sense of taste or smell. The WHO reported in the same period that the loss of taste or smell was a "less common" symptom [10]; however, a multicenter study in Europe reported 85.6% of COVID-19 patients reported loss of smell and 88.0% of patients reported loss of taste. This study also reported that infected patients present isolated olfactory or gustatory loss without any other significant symptoms [13]. As COVID-19 variants may change the COVID-19 symptoms, morbidity, and mortality landscape, it will be important to refine the type and burden of symptoms associated with COVID-19 routinely to aid screening and surveillance activities. These symptom lists may need to be customized by region to reflect the presentation of circulating variants.

In this study, one third (34.4%) of participants had an infectious or chronic comorbidity, and one in eight (12.5%) had two or more comorbidities. Over half the participants aged ≥55

years self-reported a previous diabetes diagnosis. The prevalence of comorbidities increased with age as two thirdsof adults over the age of 55 years had at least one comorbidity. The presence of a comorbidity impacted COVID-19 clinical severity. Only around 15% of adults with no comorbidity were hospitalized, compared to 40% of adults with any comorbidity and one third of overweight adults. Over half of those with two or more comorbidities were hospitalized. No death was reported among cases with no comorbidities. In this population, 34.4% of participants had at least one comorbidity–the most prevalent being overweight (24.2%) and having diabetes (10.9%), which significantly impacted their COVID-19 outcome.

Those with any comorbidity were 3.1 times more likely to have a severe outcome, as defined by hospitalization or death, compared to those patients with no comorbidity, and those participants with two or more comorbidities were 6.4 times more likely to have a severe outcome. These estimates are largely congruent with the literature from that period [14]. In a study that took place in China during December, 2019 –January, 2020, the hazard ratio for a composite outcome variable (which included admission to an intensive care unit, invasive ventilation or death) was 1.79 among patients with at least one comorbidity and 2.59 among patients with two or more comorbidities [15]. In this FFX investigation, two of the three deaths were in patients with diabetes, consistent with the literature reporting significant increased odds of death (2.78) among patients with diabetes [16]. No death was reported among cases with no comorbidities, and the 2.0% CFR was consistent with national 1.9% [9] and global reporting of 2.7% [10]. Given that over half of participants aged $\geq$ 55 years self-reported having diabetes, and that 30% of PLHIV between 25 and 44 years were not on ART, this investigation highlights the importance of integrating services to optimize every opportunity to test, refer, or reconnect participants to care.

In part, Kisumu County was selected for this investigation because of the other endemic diseases in the region–including HIV, TB and malaria. This investigation enrolled 17 people who are living with HIV (PLHIV)–one participant was first diagnosed through this study. Among cases living with HIV aged 25–44 years (n = 16), three participants were not currently on antiretroviral treatment (ART), signaling an important potential for integrating HIV screening into research studies and include linkages to care. Patients who had IgM antibodies for other infectious pathogens turn negative by PCR are presumed to have had exposure to those infections, but testing was conducted late enough in the course of disease that it missed the window when PCR would have been detectable.

In this investigation, slightly over one-third of the cases were <15 years old and nearly 50% of contacts were under 15, representing the young population structure in Kenya. Within households, children and adults appeared equally likely to acquire COVID-19. This finding is novel, and different from a systematic review and meta-analysis of transmission dynamics that reported that the risk of household transmission in adults is about 3-times higher than that in children [16]. It is possible that cases were more comfortable naming their children as contacts rather than other adult friends and co-workers resulting in an over-representation of children in this investigation, and an under-representation of adult contacts which could lead to an under ascertainment of secondary adult cases. It is also possible that the young median age in Kenya is reflected in this finding. The same systemic review and meta-analyses noted above reported that the risk of infection to household contacts is 10 times higher than other contacts [16]. Given this, along with effective IPC activities being implemented in health facilities in Kenya, the lack of transmission from health care workers to other health care workers is not surprising–and is encouraging.

This study has several inherent strengths. Given the delay between the first case in Kenya and the first case identified in Kisumu County, the FFX investigation team was able to enroll the first known COVID-19 case in Kisumu County and follow the initial spread of COVID-19

in the study area. The close collaboration between the KCDoH, the FFX investigation team, and the laboratory team allowed for an efficient and coordinated approach to enroll consecutive cases. This investigation also had limitations, including enrolling far fewer close contacts than expected. Early in the national Kenya COVID-19 pandemic, the MoH was tracing 20 contacts for each identified case. In this investigation, just a few months later, we observed increased COVID-19 stigma and reluctance to quarantine in mandated isolation centers, and therefore encountered significant resistance in eliciting contacts from cases. As a result, it's likely that cases enrolled in this investigation have contacts whom they did not name, especially adult contacts not in their households. There was also significant case and contact mobility out of the county which interfered with follow up. Finally, enrollment refusal and specimen collection refusal during the second visit resulted in incomplete follow-up for some contacts.

This investigation provided the opportunity to study the transmission of SARS-CoV-2 in Kisumu County starting with the first known case in June 2020, which started with truck drivers and public transport drivers, expanded to office workers, and then was transmitted in the community. Although onward transmission from cases to contacts was low, the majority of transmission occurred within households. Consistent with findings from other studies, those with infectious and non-communicable comorbidities had increased likelihood of severe disease or death.

## Conclusions

Our findings underscore the importance of several considerations in planning for future outbreaks pandemics. While our symptom prevalence aligned with what was being observed in other countries, it was ten times that reported nationally; small rapid assessments like this study which incorporate a follow-up period are an important addition to routine laboratory surveillance. Additionally, our prevalence and burden of specific symptoms was different than global and national reporting; these rapid assessments with follow-up can provide contextual insight into how new pathogens act in a local outbreak. Our assessment also provided clear insight into the association between chronic disease and COVID-19 outcomes at a time when this relationship wasn't widely known. The validation of these findings globally since 2020 underscores how small rapid assessments in the early days of an epidemic are valid and may be used to tailor infection and prevention control measures to those at greatest risk. In the current context of the ongoing COVID-19 pandemic with its changing landscape of SARS-CoV-2 variant strains, a review of common symptoms in our context, how they are used for screening and monitoring disease progression, and how symptoms and comorbidities are associated with poor outcomes, is overdue.

## Supporting information

**S1 Data. Dataset for primary and secondary cases.**
(DTA)

**S2 Data. Dataset for contacts of primary and secondary cases.**
(DTA)

## Author Contributions

**Conceptualization:** Beth A. Tippett Barr, Amy Herman-Roloff, Elizabeth Bukusi, Elizabeth Oele, Jayne Lewis-Kulzer, Elizabeth Hunsperger, Rachel H. Joseph, Peninah Munyua, Victor Akelo, Craig R. Cohen.

**Data curation:** Beth A. Tippett Barr, Margaret Mburu.

**Formal analysis:** Beth A. Tippett Barr, Margaret Mburu, Jayne Lewis-Kulzer, Rachel H. Joseph.

**Funding acquisition:** Amy Herman-Roloff, Peninah Munyua, Marc Bulterys.

**Investigation:** Beth A. Tippett Barr, Pamela M. Murnane, Norton Sang, Elizabeth Bukusi, Elizabeth Oele, Jayne Lewis-Kulzer, Clayton O. Onyango, Francesca Odhiambo, Rachel H. Joseph, Peninah Munyua, Kephas Othieno, Edwin Mulwa, Victor Akelo, Craig R. Cohen.

**Methodology:** Beth A. Tippett Barr, Amy Herman-Roloff, Pamela M. Murnane, Elizabeth Bukusi, Elizabeth Oele, Albert Odhiambo, Jayne Lewis-Kulzer, Francesca Odhiambo, Rachel H. Joseph, Peninah Munyua, Kephas Othieno, Edwin Mulwa, Victor Akelo, Erick Muok.

**Project administration:** Beth A. Tippett Barr, Amy Herman-Roloff, Norton Sang, Jayne Lewis-Kulzer, Elizabeth Hunsperger, Francesca Odhiambo, Peninah Munyua, Edwin Mulwa, Charles Nzioka, Craig R. Cohen.

**Resources:** Amy Herman-Roloff, Norton Sang, Elizabeth Bukusi, Jayne Lewis-Kulzer, Peninah Munyua, Erick Muok, Marc Bulterys, Craig R. Cohen.

**Supervision:** Beth A. Tippett Barr, Amy Herman-Roloff, Pamela M. Murnane, Norton Sang, Elizabeth Bukusi, Albert Odhiambo, Jayne Lewis-Kulzer, Kephas Othieno, Erick Muok, Charles Nzioka.

**Validation:** Beth A. Tippett Barr, Margaret Mburu, Clayton O. Onyango, Elizabeth Hunsperger, Rachel H. Joseph, Kephas Othieno.

**Visualization:** Beth A. Tippett Barr, Jayne Lewis-Kulzer.

**Writing – original draft:** Beth A. Tippett Barr, Amy Herman-Roloff, Craig R. Cohen.

**Writing – review & editing:** Beth A. Tippett Barr, Amy Herman-Roloff, Margaret Mburu, Pamela M. Murnane, Norton Sang, Elizabeth Bukusi, Elizabeth Oele, Albert Odhiambo, Jayne Lewis-Kulzer, Clayton O. Onyango, Elizabeth Hunsperger, Francesca Odhiambo, Rachel H. Joseph, Peninah Munyua, Kephas Othieno, Edwin Mulwa, Victor Akelo, Erick Muok, Marc Bulterys, Charles Nzioka, Craig R. Cohen.

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
