## [Decision Letter · Decision Letter 0]

26 Oct 2021

PGPH-D-21-00645

First cases of SARS-CoV-2 infection and secondary transmission in Kisumu, Kenya

Dear Dr. Tippett Barr,

Thank you for submitting your manuscript to PLOS Global Public Health. After careful consideration, we feel that it has merit but does not fully meet PLOS Global Public Health’s publication criteria as it currently stands. Therefore, we invite you to submit a revised version of the manuscript that addresses the points raised during the review process.

We look forward to receiving your revised manuscript.

Kind regards,

Megan Coffee, MD, PhD

Academic Editor

Journal Requirements:

1. Please provide additional details regarding participant consent. In the ethics statement, please ensure you have specified what type you obtained (for instance, written or verbal, and if verbal, how it was documented and witnessed).

2.  Please provide separate figure files in .tif or .eps format only, and remove any figures embedded in your manuscript file.  If you are using LaTeX, you do not need to remove embedded figures.

3. In the online submission form, you indicated that "Data can be provided by the lead author upon request."

4. Please amend your detailed Financial Disclosure statement. This is published with the article, therefore should be completed in full sentences and contain the exact wording you wish to be published.

i). State the initials, alongside each funding source, of each author to receive each grant.

ii). State what role the funders took in the study. If the funders had no role in your study, please state: “The funders had no role in study design, data collection and analysis, decision to publish, or preparation of the manuscript.”

Additional Editor Comments (if provided):

Thank you very much for your submission. Your work entitled ""First cases of SARS-CoV-2 infection and secondary transmission in Kisumu, Kenya" explores precisely the subject matter we hope PLOS Global Health will be able to focus on. The experience of early cases of COVID in Kisumu is very important for our expanding understanding of COVID in different contexts but also the use of FFX may be particularly important looking forward to future pathogens of interest. You will see the reviewers appreciated the subject and the work done here, but would like to have further clarifications in order to publish the paper, particularly involving the communication of the statistics used. Some details can be easily clarified; some may not be and that is understandable. Further figures/graphs to better communicate the concepts were also suggested. I hope this is helpful. This work is very important and I hope we are able to publish this paper.

Reviewers' comments:

Reviewer's Responses to Questions

**Comments to the Author**

1. Does this manuscript meet PLOS Global Public Health’s publication criteria? Is the manuscript technically sound, and do the data support the conclusions? The manuscript must describe methodologically and ethically rigorous research with conclusions that are appropriately drawn based on the data presented.

Reviewer #1: Yes

Reviewer #2: Partly

2. Has the statistical analysis been performed appropriately and rigorously?

Reviewer #1: No

Reviewer #2: No

3. Have the authors made all data underlying the findings in their manuscript fully available (please refer to the Data Availability Statement at the start of the manuscript PDF file)?

Reviewer #1: Yes

Reviewer #2: Yes

4. Is the manuscript presented in an intelligible fashion and written in standard English?

Reviewer #1: Yes

Reviewer #2: Yes

5. Review Comments to the Author

Reviewer #1: The manuscript seems to interesting but overall analysis is very poor, basic descriptive statistics no test statistics in table. In addition, the relevance of paper is not clearly out and no recommendations etc

I would suggest to re-do analysis with advance analysis not only descriptive.

Reviewer #2: The authors described an investigations they conducted among the first 152 cases of laboratory confirmed SARS-CoV-2 cases in Kenya. The generally could provide useful information to public health practitioners and policy makers on the transmission cycle in some African countries. The manuscript however lack clarity in coherence and presentation of thoughts hence making it difficult to read and related the events presented. My comments are below:

Abstract

The abstract needs some additional details to bring out the findings of the work. Eg. Authors could highlight the nature of the 152 confirmed cases? How many were primary cases? How many secondary cases? Of the 248 contacts, how many were household contacts or non-household contacts. Authors should rewrite the abstract in a more clearer way.

Line 40:Authors should correct the statement -- "This study, conducted June 10–October 8, 2020, included".

Line 46: The are double brackets in the text. Authors should remove this.

Results

The authors should mention the breakdown of the 152 laboratory confirmed cases. How many of the primary cases were identified through the private laboratories?

Line 106: Definition of close contact is a bit confusing? How were authors able to ascertain the 2 meters close contacts with primary cases? Did they measure this? What was the duration of contact between primary cases and contacts? Were the cases of contact not observing any of the COVID-19 protocols? Was any of them in mask? What was the national COVID-19 regulations at the time of the study?

Line 113: Authors should include the number of patients tested and those positive or negative for the cases and contact groups? This will help bring some clarity to the follow up done and those who turn out positive.

Line 116- 120: What were the whole and serum blood samples collected used for? What laboratory analysis was performed on them? What were the immunological parameter measured? What was the clotted blood used for?

Line 143: Authors definition of severe disease is also not clear?? Was this based on a national guideline? How did authors distinguish between moderate and severe disease? How come death was considered as severe disease?

Line 159: Table 1 should be reconstructed to describe the participant characteristics by cases and contact without inclusion of gender comparison. Including gender at this stage makes it difficult to read through the results. Table 1 is also not referenced in the text. The subtotal indicated in the table should also be removed. All the levels of age, occupation and activities have the same sub-total so it not clear why authors kept repeating this.

Line 166: Figure 2 intends to describe the trends among cases. Authors could best represent this as line plots by dates. Its difficult to understand the rising cases over time for the different occupational groups.

Line 179: Authors mentioned about combination of symptoms but its not clear which symptoms they were talking about?

Line 191: Table 3: What are the various definitions for the different comorbidities mentioned?

Line 201: Table 4 seem to lack clarity? What is the odd ratio measuring? Is it the risk of hospitalisation or death? This is quite confusing and ambiguous. Authors should look at this separately as either risk of hospitalisation separately from deaths. Again doing this analysis for particular age categories is also difficult to interpret along the other variables?

Line 220: Line 220: Authors seem to show some risk factor analysis being presented but interpretation is not clear?

"those with known comorbidities were no more likely to acquire COVID-19 than those without comorbidities (OR= 1.46 [95%CI 0.42- 5.12], p=0.56)" The odd ratios are all not significant and their interpretation is not relevant not relevant in explaining the risk of infection? Authors should conduct proper risk factor analysis to clearly show the risk of infection for the contacts groups.

Line 235: Authors mentioned secondary transmission occurred in 27 contacts were identified by 12 cases??? I did not see this clearly in the results. Authors should relook at the statement and describe this well.

Line 235: Again this particular statement was not significant so its not clear why authors still considered the finding as relevant in their discussion.

Line 237: Authors did not state whether the contacts or primary cases identified were in mask or used any of the COVID-19 prevention protocols? Is it that the transmission among contracts of health workers was zero because they were in mask?

Authors need to get assistance of epidemiologist and biostatisticians to help them re-organise their data in a way that will be more meaning to readers.

6. PLOS authors have the option to publish the peer review history of their article (what does this mean?). If published, this will include your full peer review and any attached files.

**Do you want your identity to be public for this peer review?** For information about this choice, including consent withdrawal, please see our Privacy Policy.

Reviewer #1: No

Reviewer #2: No

---

## [Decision Letter · Decision Letter 1]

8 Mar 2022

PGPH-D-21-00645R1

First 152 cases of SARS-CoV-2 infection and secondary transmission in Kisumu, Kenya

Dear Dr. Beth Tippett Barr:

Thank you for submitting your manuscript to PLOS Global Public Health. After careful consideration, we feel that it has merit but does not fully meet PLOS Global Public Health’s publication criteria as it currently stands. Therefore, we invite you to submit a revised version of the manuscript that addresses the points raised during the review process.

We look forward to receiving your revised manuscript.

Kind regards,

Megan Coffee, MD, PhD

Academic Editor

Journal Requirements:

Additional Editor Comments (if provided):

Reviewers' comments:

Reviewer's Responses to Questions

**Comments to the Author**

1. If the authors have adequately addressed your comments raised in a previous round of review and you feel that this manuscript is now acceptable for publication, you may indicate that here to bypass the “Comments to the Author” section, enter your conflict of interest statement in the “Confidential to Editor” section, and submit your "Accept" recommendation.

Reviewer #2: All comments have been addressed

Reviewer #3: (No Response)

2. Does this manuscript meet PLOS Global Public Health’s publication criteria? Is the manuscript technically sound, and do the data support the conclusions? The manuscript must describe methodologically and ethically rigorous research with conclusions that are appropriately drawn based on the data presented.

Reviewer #2: Yes

Reviewer #3: Yes

3. Has the statistical analysis been performed appropriately and rigorously?

Reviewer #2: Yes

Reviewer #3: Yes

4. Have the authors made all data underlying the findings in their manuscript fully available (please refer to the Data Availability Statement at the start of the manuscript PDF file)?

Reviewer #2: Yes

Reviewer #3: Yes

5. Is the manuscript presented in an intelligible fashion and written in standard English?

Reviewer #2: Yes

Reviewer #3: Yes

6. Review Comments to the Author

Reviewer #2: Authors have addressed most of the comments raised. I do understand there might be limitations in the nature of data acquired during the early stages of the pandemic but nevertheless, this work provides some information that may be useful to the general public.

Reviewer #3: Line 94 - is there a reference to support this information?

Line 127 - you say mask mandates were not implemented, but in the rest of the manuscript, you say that wearing of masks in public was mandatory... please align all comments on mask wearing to give same meaning and interpretation

Discussion - has much of results presented in it and reads somewhat like the results section. Review discussion to present more of authors thoughts on the study findings and what next steps should look like or authors thoughts on follow up studies

7. PLOS authors have the option to publish the peer review history of their article (what does this mean?). If published, this will include your full peer review and any attached files.

**Do you want your identity to be public for this peer review?** For information about this choice, including consent withdrawal, please see our Privacy Policy.

Reviewer #2: **Yes: **Dr Michael Owusu

Reviewer #3: No

---

## [Decision Letter · Decision Letter 2]

2 Aug 2022

First 152 cases of SARS-CoV-2 infection and secondary transmission in Kisumu, Kenya

PGPH-D-21-00645R2

Dear Beth A Tippett Barr

We are pleased to inform you that your manuscript 'First 152 cases of SARS-CoV-2 infection and secondary transmission in Kisumu, Kenya' has been provisionally accepted for publication in PLOS Global Public Health.

Please note that your manuscript will not be scheduled for publication until you have made the required changes, so a swift response is appreciated. (Do make sure to correct the chikungunya numbers, as noted below).

Best regards,

Megan Coffee, MD, PhD

Academic Editor

Thank you for all of your work and tackling the challenges that come with this work.

There was one typo that should be corrected. Chikungunya numbers are listed twice: "18 (12.5%) Chikungunya: 18 (12.5%)"

Reviewer Comments (if any, and for reference):

Reviewer's Responses to Questions

**Comments to the Author**

1. If the authors have adequately addressed your comments raised in a previous round of review and you feel that this manuscript is now acceptable for publication, you may indicate that here to bypass the “Comments to the Author” section, enter your conflict of interest statement in the “Confidential to Editor” section, and submit your "Accept" recommendation.

Reviewer #2: All comments have been addressed

Reviewer #3: All comments have been addressed

2. Does this manuscript meet PLOS Global Public Health’s publication criteria? Is the manuscript technically sound, and do the data support the conclusions? The manuscript must describe methodologically and ethically rigorous research with conclusions that are appropriately drawn based on the data presented.

Reviewer #2: Yes

Reviewer #3: Yes

3. Has the statistical analysis been performed appropriately and rigorously?

Reviewer #2: Yes

Reviewer #3: Yes

4. Have the authors made all data underlying the findings in their manuscript fully available (please refer to the Data Availability Statement at the start of the manuscript PDF file)?

Reviewer #2: Yes

Reviewer #3: Yes

5. Is the manuscript presented in an intelligible fashion and written in standard English?

Reviewer #2: Yes

Reviewer #3: Yes

6. Review Comments to the Author

Reviewer #2: Authors have addressed my comments and I deem the manuscript acceptable

Reviewer #3: The authors have addressed comments from the previous revision, and I appreciate that some challenges exist with reporting in the early days of the pandemic.

7. PLOS authors have the option to publish the peer review history of their article (what does this mean?). If published, this will include your full peer review and any attached files.

**Do you want your identity to be public for this peer review?** For information about this choice, including consent withdrawal, please see our Privacy Policy.

Reviewer #2: No

Reviewer #3: No
